# Behaviors and Attitudes of Polish Health Care Workers with Respect to the Hazards from Blood-Borne Pathogens: A Questionnaire-Based Study

**DOI:** 10.3390/ijerph16050891

**Published:** 2019-03-12

**Authors:** Anna Garus-Pakowska, Mariusz Górajski

**Affiliations:** 1Department of Hygiene and Health Promotion; Medical University of Lodz, 90-752 Lodz, Poland; 2Department of Econometrics, University of Lodz, 90-214 Lodz, Poland; mariusz.gorajski@uni.lodz.pl

**Keywords:** needlestick injuries, sharp injury, health care workers, occupational exposure, risk factors, knowledge, behaviors, underreporting, hospitals, Poland

## Abstract

Blood-borne infections represent an important occupational health issue in health care settings. The aim of this study was to analyze behaviors of health care workers (HCWs) in the field of needlestick injuries (NSIs) as well as to learn about their attitudes to patients infected with blood-borne viruses. A total of 487 HCWs based at 26 hospitals in Poland completed an anonymous self-administered questionnaire in the period of October–December 2015. Data was analyzed using descriptive statistics and multiple logistic regression. Of the HCWs, 44.8% suffered superficial wounds, and 17.9% HCWs were cut deeply at least once. The most frequent causes of injuries were: rush (31.4%), unpredictable patient behavior (29%), and lack of attention (27%). The rate of underreporting NSIs was 45.2%. Males showed more than three times higher chance of not reporting injuries (odds ratio (OR) 3.495, 95% Confidence Interval (CI): 1.65–7.49). The nurses more often took off their protective gloves to make the procedure easier (*p* = 0.036). Taking off protective clothes was positively associated with long work experience (OR 1.16, 95% CI: 0.995–1.36). Recapping concerned 15.5% of doctors, 8.2% of nurses, and 11.2% of paramedics. 25.9% HCWs feared infection in the workplace, and every tenth HCW refused to help the infected patient. The longer the work experience, the greater the concern about the possibility of infection (OR 1.33, 95% CI: 0.99–1.78). Most HCWs were more cautious when dealing with an infected patient and in their opinion infected patients should be required to inform HCWs of their serological status and such information should be compulsorily transferred between different health institutions. The emphasis in the training of HCWs in the future should be on classes perfecting practical skills like paying more attention to reporting NSIs, improving occupational behaviors like avoiding needle recapping, and on the development of appropriate attitudes towards patients infected with HIV, HBV, or HCV.

## 1. Introduction

Among the more than 60 different pathogens transmitted through the bloodstream, and thus threatening workers in health care settings hepatitis B virus (HBV), hepatitis C virus (HCV), and human immunodeficiency virus (HIV) are the most common and carry a serious risk of complications. They also cause social consequences, such as stigma and discrimination, and economic consequences associated with diagnostics and treatment [1,2,3,4,5]. The risk of infection of the exposed person (from a single needlestick injury—NSI, by a contaminated needle) is estimated to range between 10–30% for HBV [6], 1.8–10% for HCV [7], and 0.3% for HIV infection [8]. WHO reports in the World Health Report 2002, that of the 35 million health-care workers, 2 million experience percutaneous exposure to infectious diseases each year. It further notes that 37.6% of Hepatitis B, 39% of Hepatitis C and 4.4% of HIV/AIDS in health care workers (HCWs) around the world are due to needlestick injuries (NSI) [9].

In Poland, there is no surveillance system regarding needlestick and sharp injuries (NSIs) but in the structure of infectious and parasitic occupational diseases among health care and social workers HBV, HCV, and Tuberculosis are the most common [10]. Tuberculosis is spread from person to person through the air but cases of blood-borne infection from NSI are also described [11]. It is important that in the epidemic chain HCW may be at risk of infection, but infected-HCW can be a source of infection for the patient [12]. Therefore, it is important to take all actions to prevent injuries to medical personnel. For this purpose, first in this manuscript, prevalence of NSIs and risk factors will be identified. Literature describes such NSIs risk factors as age, short work experience, long working hours and working in surgical or intensive care units [13]. Knowledge of the risk of infection, routes of transmission, and possible prevention is an important aspect in the development of appropriate behavior of HCWs in response to exposure to infectious material.

In psychology, the theories of reasoned action and planned behavior represent connect beliefs with behavior [14]. In accordance with the theory of reasoned action, the best basis for predicting planned behaviors is the attitudes of people towards a particular behavior and their subjective norms (SN), the behavioral intention and behavior are created from these two aspects. Subjective norms, that is, people’s beliefs about how others, whose opinion matters, will react to this behavior, are a very important factor. If we want to make someone happy, we will do something even against our normal attitudes. To increase the prognostic power of the theory Ajzen introduced a ratio of perceived behavioral control as a third determinants of behavior intentions. The theory of planned behavior assumes that perceived behavioral control is an entity’s ease or difficulty in achieving a particular behavior [15]. Understanding these three elements allows us to well predict the behavioral intentions of others that are highly correlated with their planned behaviors.

In this paper we do not study the actual psychological mechanisms leading to behaviors of HCWs, but we simply apply the random utility theory which is based on the hypothesis that every individual is a rational decision-maker, maximizing utility relative to his or her choices [16]. In our study, we described 4 improper or harmful behaviors of HCWs: taking off protective clothing (*tc_i_*), no participation in trainings (*nt_i_*), lack of NSIs reporting (*lr_i_*), and recapping (*re_i_*). Furthermore, we look for determinants of the following 4 HCWs attitudes to patients infected with blood-borne viruses: fear of infection (*fs_i_*), long-term fear (*fl_i_*), needs of introducing an information system about infected patients (*is_i_*), and indifference (*in_i_*).

A number of strategies are available for avoiding the disease burden associated with NSIs, including vaccination against HBV, post-exposure prophylaxis, reducing the number of injections and invasive procedures where appropriate, using safer devices and properly disposing of needles and other sharps [9]. Access to personal protective equipment, such as gloves, gowns, masks etc. and hand washing are non-specific methods of protection against many infectious diseases, and the described negligence in this area remind us of the constant need for training [17]. In our study we would like to draw attention to such negative behaviors as removing protective gloves while doing work at the patient, or putting on the covers for used needles.

The aim of the study was to analyze behaviors of HCWs in the field of needlestick injuries as well as to learn about their attitudes to patients infected with blood-borne viruses. We estimated the impact of knowledge on the behaviors and attitudes of HCWs. Additionally, we rated the frequency of contact of the HCWs with blood and other body fluids, circumstances that may have contributed to the injury (risk factors), and the NSIs reporting.

## 2. Materials and Methods

The present work is part of the study on exposure to infectious material of HCWs in Poland. In 2015, we sent inquiries - surveys to all hospitals in Poland (*N* = 956, as of 31 December 2014) to collect registry data on HCWs NSIs. As many as 252 hospitals sent us official register data (return index = 26.36%). Representing a total annual average of 28,051 physicians, 64,806 nurses/midwifes and 3449 paramedics. We selected 26 (10%) of these hospitals to which we sent a questionnaire in electronic form. It was a selection based on the accessibility of the respondents (courtesy, cooperation and interest of the management of hospitals). The survey was for volunteer HCWs so we did not calculate the return index. 487 completed questionnaires were returned correctly. Thus, the study was conducted in 26 selected hospitals in the period of October–December 2015 from which we obtained 487 surveys.

The hospitals were located in urban communes and in urban–rural communes. In Poland, the commune is the basic unit of local self-government. The urban commune is a community located in a city. The urban–rural commune consists of a city that is the seat of the commune authorities and the surrounding villages.

A self-administered questionnaire was designed to assess HCW’s knowledge, behaviors and attitudes towards hand hygiene and needlestick injuries.

Regarding knowledge a short test was included focusing on five statements, to which possible answers were: “true”, “false” or “I do not know”. The following questions asked were:Q1.Hand disinfection can be replaced by the use of protective gloves.Q2.In an emergency situation, the disinfection of hands is not required.Q3.Approximately 60% of HBV infections among adults in Poland are nosocomial.Q4.In the case of a single puncture by used needle, it is easier to become infected with HIV than HBV.Q5.Tuberculosis infection is possible only by droplets.

The answers to the above questions constitute five independent variables, which measure the level of HCW’s knowledge. We also aggregated the results of knowledge that defined the overall knowledge variable classified as poor (less than 3 corrected answers), fair (3 corrected answers) and good (more than 3 corrected answers). In addition, we asked about the sources of knowledge and about participation in training on post-exposure proceedings.

Regarding behaviors, we asked how often personal protection equipment is used; has it happened that the HCW had removed protective clothing (e.g., gloves) to perform the operation “more easily” with the patient; we also asked about the recapping of used needles and the reporting of NSIs.

In terms of attitudes, we asked HCWs about the fear of infection in the workplace and also about changing their own behavior under the influence of awareness of care for an infected patient. We analyzed HCWs opinions on the obligation to provide information about infected patients. We also evaluated the feelings of HCWs after NSIs. In the questionnaire the variable “soon forgot about it” meant that the HCW who had been hurt was able to do his/her job further. And the variable “felt a long-lasting fear” meant that the HCW who had been hurt interrupted his/her work on the day of the risky event, “was paralyzed”, and could not continue to “work” normally, s/he was thinking about a possible threat all the time.

To assess the frequency of exposure to infectious material in the workplace, we asked about the occupational exposure which occurred in the 12 months preceding the survey. Occupational exposure to blood-borne pathogens among HCWs includes percutaneous exposures to needles and other sharp objects, and mucocutaneous exposure (i.e., contact with intact or nonintact skin, and contact with mucous membranes) [18]. In this study “superficial wounds” were defined as “loss of epidermis only” and “deep cuts” were defined as “damage to deeper tissues like tendons, muscles, ligaments, nerves, blood vessels, or bones”. In addition, we examined the circumstances that, according to the respondents, contributed to the injury.

The reliability and validity of the survey was assessed on the basis of previous studies [19].

In the present analysis, the following sociodemographic measures were taken into account: gender, job category, work experience (in years), and place of employment (urban commune or rural and urban commune).

### Statistical Analysis

We assessed the degree of association between pairs of variables by presenting contingency tables in the first line, and we applied classical Fisher’s Exact test for count data and Pearson’s Chi-squared test of independence. Moreover, we used Goodman and Kruskal’s gamma to measure the strength of association when both variables were measured at the ordinal scale, and for nominal variables, we calculated contingency coefficients.

The above mentioned behaviors (4 dependent variables) and attitudes (4 dependent variables) of HCWs were measured by simultaneous observation of dichotomous categorical variables received from the self-administered questionnaire. The values of each dependent variable falls into one of two categories, “Yes, I confirm this behavior/attitude” or “No, I do not confirm this behavior/attitude”. To determine the main drivers (factors) of healthcare workers’ behaviors and attitudes for each indicator variable we performed logistic regression with the following 6 explanatory (controls) variables: job category ($jb_{i}$ with 3 categories: doctors, nurses or paramedics), gender ($g_{i}$), work experience ($exp_{i}$ with 4 categories: less than 5, 6–15, 16–25, >25 years), and place of employment ($pemp_{i}$ with 2 categories: urban commune or rural and urban commune), personal situation ($ps_{i}$ with 2 categories: I feel insecure at my workplace “ or “I am professionally fulfilled”) and the overall knowledge of worker ($kn_{i}$, with 3 categories poor, fair and good). For a given worker i, let $y_{i}$ be a dichotomous variable describing his/her particular behavior or attitude i.e., $y_{i}\in \left\{tc_{i},nt_{i},lr_{i},re_{i}\right\}\cup \left\{fs_{i},fl_{i},is_{i},in_{i}\right\}$, and let $U_{y_{i}}$ be the internal utility that worker i obtains from this behavior or attitude. The utility gains are given by $U_{y_{i}}=\beta_{0}+\beta ’x_{i}+\epsilon_{i}$ and depends on the above listed characteristics of the worker and her place of employment grouped into a vector $x_{i}=\left[jb_{i},g_{i},exp_{i},kni,pemp_{i},ps_{i,}\right]$, where the unobserved term, $\epsilon_{i}$, is random shock with logistic distribution. The HCW i takes the attitude or behavior ($y_{i}=1$) if $U_{y_{i}}>0$ and do not if $U_{y_{i}}<0$. The conditional probability of $y_{i}=1$ is given by $Pr\left(y_{i}=1|x_{i}\right)=\frac{1}{1+e^{-\beta_{0}-\beta ’x_{i}}}$.

To find the best model we performed the best subset regression algorithm and fitted a separate logistic regression for each possible combination of the k=1,2,3,4,5,6 predictors. We then look at all of the resulting models and identified the one that is best using the following rule: all predictors in the model are statistically significant and the value of the Akaike Information Criterion is minimized. To assess the influence of control variables on behaviors and attitudes we calculated the odds ratios.

The statistical computations were performed using R statistical software and Excel. The level of statistical significance was set at *p* < 0.05. The study protocol was approved by the Bioethics Committee of the Medical University of Lodz (Document No. RNN /163/14/KB of 11.02.2014).

## 3. Results

### 3.1. Characteristics of the Study Group

A total of 487 HCWs filled the questionnaire. About half of the study participants were female (56.9%), 44% of them were paramedics and 40% were nurses, and 31.4% had less than 5 years of employment. There was approximately an equal distribution among hospitals located in urban commune (44%) and rural/urban commune (55%), Table 1.

### 3.2. Frequency and Circumstances of Injuries

During the 12 months preceding the study, most doctors, nurses and paramedics were in contact with the infectious material through intact skin. They had less contact through damaged skin, mucous membranes and splattering onto conjunctiva. Almost every second HCW (44.8%) suffered superficial wounds, and almost every fifth HCW (17.9%) had been cut deeply at least once. More often, both the superficial and deep injuries were among nurses (respectively to the second and every fourth) (*p* < 0.05), Table 2. The most frequent causes of injuries were: rush (31.4%), unpredictable patient behavior (29%), lack of attention (27%), stressful situation requiring urgent intervention such as sudden hemorrhage or collapse (16.3%), as well as too high workload (14.3%).

### 3.3. Knowledge, Behaviors, and Attitudes

The correct answers to all five questions from the knowledge test were given by 25 doctors (32.9%), 34 paramedics (15.8%) and 18 nurses (9.2%). The most common were 4 correct answers (doctors: *n* = 31, 40.8%, nurses: *n* = 63, 32.1%, paramedics: *n* = 85, 39.5%). We rated the level of knowledge as poor for 15% of HCWs, fair for 31% of HCWs, and good for 54% of HCWs.

The distribution of correct answers to individual questions is given in Table 3. The frequency of providing correct answers did not depend on the sex, seniority, or location of the hospital. Q3, Q4, and Q5 questions were more often answered correctly by doctors (*p* < 0.05), Table 3.

Among the 487 respondents, the majority of HCWs 79.5% (*n* = 387) felt the need to increase knowledge about the possibility of getting infected in the workplace. However, simultaneously as many as 82.1% of HCWs responded that their knowledge was gained during the basic training (school/study). 68.6% HCWs participated in various refresher courses, but only less than every second (45.6%) read scientific journals. Nurses most often participated in the trainings (chi^2^ = 36.813, *p* < 0.05). For example, 25% of doctors, 22.8% of rescuers, and 10.2% of nurses never participated in the training on post-exposure proceedings.

Regarding behaviors, most HCWs used personal protective equipment such as clothes and protective gloves. Only 3.7% of employees answered that they never use protective gloves, 10.1% protective gowns, and 49.8% goggles or other face shields. However, it should be remembered that not all activities performed on the patient require the use of protective clothing. At the same time, a large number of respondents admitted that they took off protective clothing (e.g., gloves) to perform an activity “more easily” on the patient (doctors: *N* = 33, 43.4%; nurses: *n* = 116, 59.2%; paramedics: *N* = 111, 48.4%). This answer was the most common among nurses (*p* = 0.036).

On average, every tenth HCW (11.7%) was spotted for improper handling of used needles. Recapping concerned 15.5% of doctors (*n* = 12), 8.2% of nurses (*n* = 16) and 11.2% of paramedics (*n* = 24). Among them, when placing the cover on the contaminated needle, 2 doctors, 4 nurses and 4 paramedics were injured.

Most HCWs (*n* = 440, 90.3%) answered that they had always reported the fact of injury to the appropriate person responsible for keeping the register. Most often they were nurses (92.3%) and paramedics (93.5%) (*p* < 0.01). However, when asked about reporting the last NSIs that occurred in the last 12 months preceding the study, only 54.8% of HCWs reported this fact. The other half of incidents (45.2%) have not been reported anywhere. The main reason for not reporting the sharp injuries was the perception that there was no such need. The variables such as seniority, place of employment and knowledge did not have any significant impact on the attendance.

Regarding attitudes, every fourth HCWs (25.9%) feared infection in the workplace. Nurses had the most concerns (37.2%, *p* < 0.01), women (31.0%, *p* < 0.01), and HCWs with the shortest work experience (31.4%, *p* < 0.01). After the last injury, 35.2% of HCWs felt fear/anxiety but soon forgot about it. However, 16.3% of HCWs felt a long-term fear for their own health. Every third HCW (30.8%) did not feel anything special, arguing that NSIs are an inseparable part of the work of medical personnel. A small percentage of HCWs (5.3% doctors, 6.6% nurses and 1.4% paramedics) did not exercise greater caution in dealing with a patient known to be infected with HIV, HBV, or HCV. HCWs with shorter work experience mostly changed their behavior (83% from <5 years work experience and 78.9% HCWs from 6–15 years of work experience, *p* = 0.002). The HCWs from rural–urban communes changed their attitudes more often (*p* = 0.052). 93.4% of nurses, 89.5% of doctors and 87.0% of paramedics have never refused to perform the examination, surgery, or care of a patient with an infection. However, the others feared personal infection and they refused to help the infected patient at least once. Most HCWs (69.4%) would like patients to be required to inform medical personnel that they are infected with blood-borne viruses. Even more HCWs (78.2%) see the need to introduce in Poland an obligatory system of transmitting information about infected patients between different health care units (throughout the health care service system).

### 3.4. The Influence of Knowledge and Other Variables on Behaviors and Attitudes

Multivariate logistic regression analysis of the odds (OR) for behaviors and attitudes in relation to the potential risk factors listed above is presented in Table 4.

Regarding behaviors, taking off protective clothes to make it “easier” to take action with the patient was positively associated with long work experience (odds ratio (OR) 1.16, 95% confidence interval (95% CI): 0.995–1.36), and negatively associated with knowledge (OR 0.75, 95% CI: 0.58–0.96). Recapping was positively associated with knowledge (OR 7.74, 95% CI: 1.70–13.15), and negatively associated with job category (OR 0.15, 95% CI: 0.01–0.78) and place of employment (OR 0.21, 95% CI: 0.05-0.73). Males showed more than three times higher chance of not reporting injuries (OR 3.495, 95% CI: 1.65–7.49).

Regarding attitudes none of the variables significantly affected the possibility of refusing treatment/care to an infected patient. Concern about the possibility of getting infected in the workplace was positively associated with long work experience (OR 1.33, 95% CI: 0.99–1.78). Long-term fear after NSIs was also positively associated with knowledge (OR 2.28, 95% CI: 1.29–4.43). Males were less likely to experience such long-term fear (OR 0.38, 95% CI: 0.13–1.12).

## 4. Discussion

Health care workers are at common risk of occupational exposure to blood and other potential infectious material. In the study Martins et al. 65% of employees of a selected hospital in Portugal reported having experienced at least one NSSI in the last 5 years [20]. Most of the events, similar to ours and other studies (Polish and international) [21,22,23,24,25] concerned nurses. It should be noted that nurses are the most numerous professional group among medical employees, they perform the most treatments and usually have direct contact with patients.

The risk of acquiring the blood-borne diseases through occupational exposure depends on the number of injuries, prevalence of BB infections in the patient population and probability of a percutaneous injury transmitting blood-borne viruses [26]. With increasing rate of national and international hepatitis B, hepatitis C, and HIV these are risks that HCWs cannot afford to take. In our study perceived causes of injury were: rush (31.4%), unpredictable patient’s behavior (29%), and lack of attention (27%). In the study by Salzer et al. [27] time pressure and lack of experience were the most frequent causes of NSIs, while in other studies the main reason for occupational exposure was a sudden movement of the patient during the procedure [28]. The same reasons as in our study were obtained by Bećirević et al.: being in rush, patient’s unpredictable reaction and decrease in concentration [29].

The knowledge of medical personnel should be considered insufficient. The level of knowledge of HCWs has been the subject of many studies but little is known about whether knowledge/lack of knowledge about the possible risk of infection can affect behaviors/attitudes [22,30]. In our study nurses showed the weakest knowledge. Interestingly, nurses most often participated in trainings and the least often presented negative behaviors in the recapping. At the same time, they usually took off their protective clothing to perform some procedures on the patient. Most often, they also reported NSIs. In our opinion, negative behavior may have resulted from the nature of the work rather than from lack of knowledge. Nurses have the most frequent contact with the patient, they are more likely to perform invasive procedures, so they will also have the biggest chance to remove protective clothing, e.g., protective gloves to "facilitate" their work. Such behaviors may result from excessive workload, in a hurry, and these HCWs risk factors have been indicated as the most common causes of injury. The training deals with the subject of recapping, and almost always reminds us of the obligation to report injuries. In this matter, knowledge influenced the improvement of behavior.

This study revealed that information on 45.2% of injuries was not entered into an official register. This is close to the results of some studies (i.e., 51% in the survey by Makary et al.) [31], but less than in others (i.e., 66.1% in the study by Cui et al. and 60.2% in the study by Jahangiri et al.) [32,33]. Most common reason for failure to report the incidents of NSIs were lack of time and heavy clinical schedule, as well as perception of low risk of infection [33,34] which is close to our results, in which part of HCWs in its own assessment did not see the need to report NSIs. This is still a gap for enhanced training. HCWs should be aware that reporting an injury is highly important and adjust their behavior accordingly.

Medical personnel suffer from anxiety and emotional distress following NSIs [35]. In our study, logistic regression showed that people with better knowledge are more afraid of their own health after being hurt and this fear lasted longer. The reason is unclear, but it may be related to the fact that people with better knowledge are aware of the possible dangers of wounds contaminated with a needle. However, the long-term fear was also positively correlated with the practice of nurse and rescuer, who at the same time showed lower knowledge in the knowledge test. In these cases, the fear may be due to the generally more frequent contacts of employees of these professions with infectious material. One should also take into account the possibility that the measurement of knowledge was imprecise because the test of knowledge covered only 5 questions. Probably fear for one’s own health may affect better wound reporting in the paramedics and nurses group. Men in our study were less afraid, and at the same time reported less frequent incidents of injuries. Similarly, with doctors. It is quite understandable when we are not afraid of something, we do not see the need to report. In the research by Jahić et al. also physicians were significantly less likely to report exposure incidents than other staff [36]. The vast majority of HCWs are more cautious when dealing with an infected patient. Fortunately, the cases of refusal to provide assistance to patients infected with HBV, HCV, or HIV were rare, and were not subject to any risk factors. These are results similar to Ishimaru et al. in which the majority of nurses expressed a willingness to care for patients infected with HIV, HBV or HCV [37].

One of the major alarming observations in the current study is that most HCWs believed that patients with HIV should be required to inform physicians of their status. Moreover, they think that such information should be compulsorily transferred between different health institutions. In another Polish study, Gańczak et al. found insufficient knowledge of hospital staff in surgical wards, in which 40% of HCWs were in favor of moving away from performing surgical procedures if the HCW was infected with HBV or HCV, and 42.6% with respect to HIV infected HCWs. In the same research 16.2% HCWs stated that infected surgeons should disclose their HBV, HCV, or HIV serostatus [38]. HCWs should be aware of the risks, but above all they should have knowledge about prevention methods that effectively protect against infection. Although HBV infection can be prevented by vaccination, there is no effective vaccine for HCV and HIV. Before other infections, strict compliance with the universal precautions is important. In our study the vast majority of the HCWs were afraid of being infected during work, and almost all admitted that they performed medical procedures more carefully with patients they knew to be infected with HIV, HBV, or HCV. This may indicate a lack of awareness that absolutely every patient should be treated as a potential source of infection. The use of personal protective equipment for operations performed on the patient and carefully carrying out all procedures should not be dependent on the results of the patient’s tests for markers of blood-borne virus infections, due to the possibility of patients being in the serological window or low sensitivity of the tests. The patient himself is not obliged to inform that he/she is infected; therefore, in a job where there is a possibility of contact with potentially infectious material, the same safety precautions must be used for all patients.

HCWs’ knowledge about blood-borne infections and their prevention (hand hygiene) is insufficient but in our opinion, most behaviors and attitudes depended more on the type of profession than on knowledge. Better knowledge did not influence all behaviors; doctors showed better knowledge and this affected a smaller feeling of fear and less frequent NSIs reporting. Changing behavior and attitudes under the influence of the awareness of care for an infected patient is a sign of fear, which is also mentioned by the HCWs themselves. Knowledge did not significantly change these attitudes. Perhaps this is due to the fact of textbook teaching in the Polish education system. This justifies the poorer result of the knowledge test among nurses despite the relatively more frequent participation in training. Perhaps the courses and practical experience gained contribute to better nurse behavior, for example recapping. This issue requires more in-depth research. It can also imply for decision makers on medical education the need to move from theoretical instruction to the emphasis on practice. We look forward to wide participation in this discussion.

### Limitations

Due to the number of surveys we have obtained, we can not generalize results for the whole country, and the entire population of hospital HCWs. Also the measure of knowledge may be imprecise due to the small number of questions in the test. This issue requires further research. For example, the test of knowledge could be extended with questions carefully checking whether the respondent knows what to do after the exposure, or whether he knows what health risks can result from this exposure.

## 5. Conclusions

The emphasis in the training of HCWs in the future should be on classes perfecting practical skills like paying more attention to reporting NSIs, improving occupational behaviors like avoiding needle recapping, and on the development of appropriate attitudes towards patients infected with HIV, HBV, or HCV in Poland, as elsewhere.

## Figures and Tables

**Table 1 ijerph-16-00891-t001:** Demographic characteristics of health care workers, *N* = 487 (%).

Demographic Characteristics		Number (%)
Gender	Male	210 (43.1)
	Female	277 (56.9)
Job Category	Doctors	76 (15.60)
	Nurses	196 (40.25)
	Paramedics	215 (44.15)
Work Experience (in years)	<5	153 (31.4)
	5–15	128 (26.3)
	16–25	89 (18.3)
	>25	117 (24.0)
Place of employment	Urban commune	215 (44.15)
	Rural and urban commune	272 (55.85)

**Table 2 ijerph-16-00891-t002:** Frequency of health care workers (HCWs) contacts with potentially infectious material within 12 months preceding the study.

Exposure Type	Job Category	Frequency of Exposure Within the Last 12 Months							
		Never		Once		At Least a Few Times		Statistical Significance	
		*N*	%	*N*	%	*N*	%	Chi^2^	*p*-Value
Through intact skin	Doctors	18	23.7	16	21.1	42	55.2	14.394	0.072
	Nurses	56	28.6	20	10.2	120	61.2		
	Paramedics	57	26.5	34	15.8	124	57.7		
	Total	131	26.9	70	14.4	286	58.8		
Through damaged skin	Doctors	59	77.6	9	11.8	8	10.5	28.106	<0.05
	Nurses	119	60.7	21	10.7	56	28.5		
	Paramedics	153	71.2	26	12.1	36	16.7		
	Total	331	68.0	56	11.5	100	20.5		
Through mucous membranes	Doctors	57	75.0	7	9.2	12	15.8	14.894	0.061
	Nurses	118	60.2	17	8.7	61	31.1		
	Paramedics	144	67.0	23	10.7	48	22.3		
	Total	319	65.5	47	9.7	121	24.9		
Splattering onto conjunctiva	Doctors	45	59.2	17	22.4	14	18.4	33.216	<0.05
	Nurses	110	56.1	34	17.3	52	26.6		
	Paramedics	164	76.3	28	13.0	23	10.7		
	Total	319	65.5	79	16.2	89	18.3		
Superficial injury	Doctors	43	56.6	17	22.4	16	21.0	51.054	<0.05
	Nurses	90	45.9	39	19.9	67	34.3		
	Paramedics	136	63.3	60	27.9	19	8.9		
	Total	269	55.2	116	23.8	102	21.0		
Deep injury	Doctors	62	81.6	11	14.5	3	3.9	26.618	0.001
	Nurses	147	75.0	20	10.2	29	14.8		
	Paramedics	191	88.8	16	7.4	8	3.7		
	Total	400	82.1	47	9.7	40	8.2		

Chi^2^—chi-square test of independence.

**Table 3 ijerph-16-00891-t003:** Distribution of the responses provided by the HCWs during the test of knowledge.

Question	Correct answer *N* (%)									
	Doctors		Nurses		Paramedics		Total		Statistical Significance	
	*N*	%	*N*	%	*N*	%	*N*	%	Chi^2^	*p*-Value
**Q1.** Can use of gloves replace disinfection of hands?	73	96.05	188	95.92	210	97.67	471	96.72	1.12	0.57
**Q2.** Is disinfection of hands necessary in emergency situations?	51	67.11	119	60.71	146	67.91	316	64.89	2.52	0.28
**Q3.** Do you agree that 60% of HBV infections in Poland is connected with health care?	56	73.68	112	57.14	143	66.51	311	63.86	7.66	0.02
**Q4.** Is it more likely to become infected with HIV than with HBV as a result of single needlestick injury with a contaminated needle?	71	93.42	154	78.57	169	78.61	394	80.90	9.13	0.01
**Q5.** Is infection with tubercule bacillus possible solely through droplet infection?	57	75	62	31.63	95	44.19	214	43.94	41.82	0.00

Chi^2^—chi-square test of independence.

**Table 4 ijerph-16-00891-t004:** Multivariate logistic regression analysis of odds ratio (OR) for behaviors and attitudes of HCWs in relation to potential risk factors.

Variables	Odds Ratios with 95% Confidence Interval (CI) from Logistic Regression Models							
	Behaviors, OR (95% CI)				Attitudes, OR (95% CI)			
	No Participation in Trainings	Removing Protective Clothing to “Make It Easier” to Perform the Action at the Patient	Recapping	Lack of NSIs Reporting	Fear of Infection in the Workplace	After NSIs, I Did Not Feel Anything Special	Long-Term Fear for Their Own Health After NSIs	The Need to Introduce an Information System About Infected Patients
Job category (nurses)	0.465 ** ^(1)^ (0.22–1.01)	-	0.145 * ^(2)^ (0.01–0.78)	-	2.929 *** ^(1)^ (1.35–6.35)	-	4.040 * ^(1)^ (0.99–27.67)	-
Job category (paramedics)	0.585 ** ^(1)^ (0.31–1.14)	-	-	0.241 ***^(2)^ (0.11–0.53)	2.639 ***^(1)^ (1.28–5.41)	-	7.158 ** ^(1)^ (1.83–48.25)	2.849 *** ^(2)^ (1.78–4.68)
Gender (Male)	-	-	-	3.495 *** (1.65–7.49)	2.095 ** (1.08–4.09)	-	0.378 * (0.13–1.12)	-
Work experience (increasing seniority)	0.621 *** (0.46–0.82)	1.163 * (0.99–1.36)	-	-	1.327 * (0.99–1.78)	-	-	-
Place of employment (rural and urban commune)	-	-	0.209 ** (0.05–0.73)	-	-	-	-	0.679 * (0.43–1.06)
Personal situation (I am professionally fulfilled)	0.506 *** (0.309, 0.826)	-	-	-	0.482** (0.251, 0.878)	2.858 *** (1.485, 5.839)		
Knowledge (increasing level of knowledge)	-	0.747 ** (0.58–0.96)	7.74 ** (1.70–13.72)	-	-	0.717 * (0.48–1.06)	2.281 *** (1.29–4.43)	-

Note: * *p* < 0.1; ** *p* < 0.05; *** *p* < 0.01; (1) ref. = doctors; (2) ref. = others.

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
