# Peer review of "Behaviors and Attitudes of Polish Health Care Workers with Respect to the Hazards from Blood-Borne Pathogens: A Questionnaire-Based Study"

_ijerph, 2019, doi:10.3390/ijerph16050891_

Round 1

Reviewer 1 Report

I appreciated the opportunity to review the revised manuscript. I did not see a response to the reviewer that outlined the response to each feedback, so I am primarily using the red track changes and comparison of the revised manuscript to my original notes to make that judgement. 

The authors sufficiently addressed my concerns about describing the measure in greater detail and clarity of the regression model. I would still like to see them address the potential effects of data being nested within hospitals. I''m pasting my original comment below related to that. 

The data are nested within units, within departments, and within hospitals. When the nested nature of data are not properly accounted for, we risk violating important assumptions of the regression based approach: A review article is provided below. Researchers usually use the intraclass correlation coefficient (ICC) to test whether the data are clustered and if multilevel modeling approaches are necessary.

Klein, K. J., & Kozlowski, S. W. (2000). From micro to meso: Critical steps in conceptualizing and conducting multilevel research. Organizational research methods, 3(3), 211-236.

Overall, I felt that the authors adequately addressed my concerns. 

Author Response

Dear Reviewer,

Thank you that you again wanted to read the manuscript, and for your time.
Please accept our answer below.

“I appreciated the opportunity to review the revised manuscript. I did not see a response to the reviewer that outlined the response to each feedback, so I am primarily using the red track changes and comparison of the revised manuscript to my original notes to make that judgement. 

The authors sufficiently addressed my concerns about describing the measure in greater detail and clarity of the regression model. I would still like to see them address the potential effects of data being nested within hospitals. I''m pasting my original comment below related to that. 

The data are nested within units, within departments, and within hospitals. When the nested nature of data are not properly accounted for, we risk violating important assumptions of the regression based approach: A review article is provided below. Researchers usually use the intraclass correlation coefficient (ICC) to test whether the data are clustered and if multilevel modeling approaches are necessary.

Klein, K. J., & Kozlowski, S. W. (2000). From micro to meso: Critical steps in conceptualizing and conducting multilevel research. Organizational research methods, 3(3), 211-236.

Overall, I felt that the authors adequately addressed my concerns.”

The study was conducted in 26 selected hospitals in the period of October - December 2015 from which we obtained 487 anonymous surveys. Unfortunately, our dataset does not contain information about specific medical units. Thus, we are not able to calculate the infraclass correlation coefficient (ICC), nor apply logistic regression with robust clustered standard errors. Nevertheless, we believe that behaviors and attitudes of HCWs are independent from particular medical units. Furthermore, in the logistic regressions we apply the Eicker-Huber-White robust estimators of covariance matrices of parameters. As a result, all our main conclusions have remained unchanged.

Thank you.

Reviewer 2 Report

Dear Authors, Good Work!

Here are just small corrections:

lines 26 and 362: replace "pay" with "paying"

line 38: insert acronym (NSI)

line 43: replace "needlestick injuries" with "NSI"

line 47:  replace "needlestick injuries" with "NSI"

line 89: replace "healthcare workers" with "HCW"

line 126: replace "his" with "his/her"

line 127: replace "his" with "his/her"

line 128: replace "he" with "s/he"

line 155: replace "3" with "4" (4 different work experiences, right?)

line 159: replace "his" with "his/her"

line 179: delete "(31.4%)"

line 179: replace "were" with "was"

line 351: replace "about" with "on" 

Author Response

Dear Reviewer,

Thank you for re-reading the manuscript and notes. We have carried out all the suggested changes.

Here are just small corrections:

lines 26 and 362: replace "pay" with "paying" - we replaced

line 38: insert acronym (NSI) – we added

line 43: replace "needlestick injuries" with "NSI" - we replaced

line 47:  replace "needlestick injuries" with "NSI" – we replaced

line 89: replace "healthcare workers" with "HCW" – we replaced

line 126: replace "his" with "his/her" - we added

line 127: replace "his" with "his/her" - we added

line 128: replace "he" with "s/he" – we added

line 155: replace "3" with "4" (4 different work experiences, right?) – yes, of course, thank you

line 159: replace "his" with "his/her" – we replaced

line 179: delete "(31.4%)" – we deleted

line 179: replace "were" with "was" – we replaced

line 351: replace "about" with "on"  - we changed

Thank you.

Reviewer 3 Report

The manuscript has a number of weaknesses which need to be considered. I suggest to reconsider the methods used, as well as statistical analyses which have been made.

1.     The abstract needs to be improved.

The abstract is missing key information to understand the study design; i.e. the sampling procedure, the response rate. “A total of 487 HCWs based at several hospitals in Poland…” “Several” is not a precise term, the authors should specify how many hospitals were invited. There is no information about the response rate. The results are presented in the sloppy way and are quite messy. In some parts the authors present numbers which refer to all HCWs, in some others - to specific job categories, e.g. “The rate of underreporting NSIs was 45.2%. The nurses more often took off their protective gloves to make the procedure easier (p=0.036)” and further “Recapping concerned 15.5% of doctors, 8.2% of nurses and 11.2% of paramedics. 25.9% HCWs feared infection in the workplace”. Furthermore, there is an information that data were analysed using multiple logistic regression, however this is not presented in the results section. I could not find any results of logistic regression in the abstract. Instead, there are some general statements, such as: “every tenth HCW refused to help the infected patient” (in which way? when?) and further “Most HCWs were more cautious when dealing with an infected patient ( how many? What does “more cautious“ mean?) and in their opinion infected patients should be required to inform HCWs of their serological status and such information should be compulsorily transferred between different health institutions”.

2.     Methods have to be based on adequate sampling, power calculations and should use an adequate study instrument

-          The authors have not explained how many HCWs were invited and how many agreed to participate. If the study was conducted among HCWs which were not randomly selected, the results may even not be generalizable to all HCWs at the selected hospitals. This should be clearly stated in the Limitations section.

-          The authors should illustrate the recruitment procedure in an additional figure (a graph which illustrates the sampling procedure).

-          The study lacks a sample size (or power) determination.

-          The authors should explained if the questionnaire was it pilot tested. This could probably eliminate some methodological errors related to the study instrument

-          In the questionnaire the variables "soon forgot about it" "felt a long-lasting fear", "was paralyzed" are unprecise. What does “soon forgot about it” mean? Few seconds? Few minutes? Few days? What were the criteria to differentiate between “soon forgot about it” and “felt a long term fear”? As these two might overlap this could not be taken into account as a measure of fear. In the same vein, what does “a HCW did not feel anything special” mean? This is an unprecise term and cannot be used as a variable. Furthermore, such unprecise questions were used in the questionnaires sent electronically. Therefore, respondents did not have any chance to discuss those unprecise terms with the research team.

-          The Authors state: A self-administered questionnaire was designed to assess HCW’s knowledge, behaviors and attitudes towards hand hygiene and needlestick injuries “. The question remains why 2 questions were chosen by the Authors to assess knowledge about hand hygiene and 2 others about theneedlestick injuries. Knowledge questions seem to be oriented to different issues. Such an instrument is not applicable for the purpose of this study and may lead to erroneous conclusions. I would suggest to delete knowledge-related issues from the analyses.

-          The explanation: “hospitals were located in urban communes and in urban-rural communes. In Poland, the commune is the basic unit of local self-government. The urban commune is a community located in a city. The urban-rural commune consists of a city that is the seat of the commune authorities and the surrounding villages” is confusing and is not related to the type of the hospital.

3.     The Results section

-          Table 4 is not informative and is messy. It is not clear why some variables were taken separately e.g. “nurses” and “paramedics”, some others were  not, some are described in details e.g. “gender (male)”, some others are not e.g. “Work experience”. The authors should keep the same approach regarding all variables used in the analysis.

-          The variable “knowledge” should be deleted from the logistic regression analysis.

Author Response

Dear Reviewer,

Below are our responses to comments.

1.     The abstract needs to be improved.

The abstract is missing key information to understand the study design; i.e. the sampling procedure, the response rate. “A total of 487 HCWs based at several hospitals in Poland…” “Several” is not a precise term, the authors should specify how many hospitals were invited. There is no information about the response rate. The results are presented in the sloppy way and are quite messy. In some parts the authors present numbers which refer to all HCWs, in some others - to specific job categories, e.g. “The rate of underreporting NSIs was 45.2%. The nurses more often took off their protective gloves to make the procedure easier (p=0.036)” and further “Recapping concerned 15.5% of doctors, 8.2% of nurses and 11.2% of paramedics. 25.9% HCWs feared infection in the workplace”. Furthermore, there is an information that data were analysed using multiple logistic regression, however this is not presented in the results section. I could not find any results of logistic regression in the abstract. Instead, there are some general statements, such as: “every tenth HCW refused to help the infected patient” (in which way? when?) and further “Most HCWs were more cautious when dealing with an infected patient ( how many? What does “more cautious“ mean?) and in their opinion infected patients should be required to inform HCWs of their serological status and such information should be compulsorily transferred between different health institutions”.

 According to the Instructions for Authors “The abstract should be a total of about 200 words maximum. The abstract should be a single paragraph and should follow the style of structured abstracts, (…) Methods: Describe briefly the main methods(…),  Results: Summarize the article's main findings; and  Conclusion: Indicate the main conclusions or interpretations. The abstract should be an objective representation of the article: it must not contain results which are not presented and substantiated in the main text and should not exaggerate the main conclusions”. [https://www.mdpi.com/journal/ijerph/instructions].  In our opinion, our abstract is written in accordance with the rules. Some results are given as a whole, others vary due to variables, which is what other authors do in their publications.
In the summary, we indicate the main results, we apologize but in 200 words we will not fit the entire methodology and all results - from this is a manuscript, and the summary is to encourage reading it.

Similarly, there is no place in the summary to explain all the comments made by the reviewer.
We have already explained both in the answer and in the article that the planned study does not present itself “sampling procedure, and the response rate”.  We will explain it again below.

We changed “A total of 487 HCWs based at several hospitals in Poland…” – “several” – to “26”.

We corrected the summary in accordance with the previous recommendation of the reviewer. The conclusion corresponds to the assumed objectives of the study and results.

Additionally, in the summary, we added the results of the regression model (in our opinion the most important).

2.     Methods

-          The authors have not explained how many HCWs were invited and how many agreed to participate. If the study was conducted among HCWs which were not randomly selected, the results may even not be generalizable to all HCWs at the selected hospitals. This should be clearly stated in the Limitations section.

Sample section process:

Population source: HCWs from all 956 hospitals in Poland;

Reference population: HCWs from 252 hospitals who sent us official register data (return index=26.36%)

Number of HCWs in that 252 hospitals: 28,051 physicians, 64,806 nurses/midwifes and 3,449 paramedics.

That are 31.72% of all physicians in Poland, 29.48% of nurses and 30.3% of paramedics.

Then, we sent a questionnaire to 26 hospitals (10% of that which sent us registers data). Selection of the sample based on the accessibility of the respondents – [Babbie E. “The Practice of Social Research” 9th edition. 2001 Wadsworth/Thomson Learning].

Comments:

It is difficult to say what is the return index at the level of the number of employees. We know what was the number of employees in 252 hospitals - data were given to us by hospitals.

However, we sent the questionnaires to 26 selected hospitals on the basis of availability [Babie E. 2001] and at this stage we did not interfere in numbers.

The more so because the survey was directed to the willing and such a sample selection is not representative in principle. Therefore, calculating the return index in such a study is unfounded. It does not bring anything. And it is not possible at all.

A similar situation occurs when the researcher places a survey on the website - it is not possible to determine the return index.

From the beginning, we write that the sample is not random, and that we will not generalize the results on the entire population. The description of the selection is included in the methodology, lines 94-99; restrictions are in the limitations section, lines 357-358.

-  The authors should illustrate the recruitment procedure in an additional figure (a graph which illustrates the sampling procedure). Of course, you can graphically illustrate what we described above and in the manuscript, but this will not bring anything new to the text - for the reasons described above. We did not draw the respondents. They volunteered themselves.

- The study lacks a sample size (or power) determination. A sample size is calculated in studies which results are generalized to the entire population. Our study did not have the characteristics of representativeness as we emphasized in the text (Limitations section). We believe that lack of representativeness does not diminish the value of the research and the goals set.

-The authors should explained if the questionnaire was it pilot tested. This could probably eliminate some methodological errors related to the study instrument

We wrote about it both in the article and in our response. “The reliability and validity of the survey was assessed on the basis of previous studies [19]  - lines 141.

We have also published other works based on the same survey, ex.

Garus-Pakowska, A., Górajski, M., Szatko, F.Knowledge and attitudes of dentists with respect to the risks of blood-borne pathogens - a cross-sectional study in Poland.  Int J Environ Res Public Health
2017 : Vol. 14, nr 1, s. pii: E69”

-          In the questionnaire the variables "soon forgot about it" "felt a long-lasting fear", "was paralyzed" are unprecise. What does “soon forgot about it” mean? Few seconds? Few minutes? Few days? What were the criteria to differentiate between “soon forgot about it” and “felt a long term fear”? As these two might overlap this could not be taken into account as a measure of fear. In the same vein, what does “a HCW did not feel anything special” mean? This is an unprecise term and cannot be used as a variable. Furthermore, such unprecise questions were used in the questionnaires sent electronically. Therefore, respondents did not have any chance to discuss those unprecise terms with the research team.

In this question only 1 answer was possible, so we think that the variables are disjoint; the answers did not overlap.

In addition, it had not happened that any respondent marked two or more answers in this question.

Surveys are subjective and burdened with a lack of credibility, in the sense that we are never sure whether the respondent has written the truth. But this is a feature of questionnaire research. Nevertheless, "polls are a great tool for measuring attitudes and views"  [Babie E. 2001]

What's more, there are surveys based on questionnaires in which the respondent has the answers for example: 1. I totally disagree, 2. I rather disagree 3. I rather agree, 4. I totally agree.
Another type of research is those in which the respondent has to mark their opinions on a scale of, for example, 1-5 or 1-10. In these surveys, the results are also based on subjective estimates of respondents or / and may seem imprecise.

 [ex. Rybacki M. et al. Work safety among Polish health care workers in respect of exposure to bloodborne pathogens. Med. Pr. 2013, 64, 1-10. DOI: 10.13075/mp.5893/2013/0001].

The survey has already been carried out several times and the results have been published. The question was a one-time choice. Nobody answered several questions in this question. No respondent reported comments. Always in our research, the questionnaire is annotation "in case of doubt or questions, please contact us" and we write, for example, a telephone number.

-          The Authors state: A self-administered questionnaire was designed to assess HCW’s knowledge, behaviors and attitudes towards hand hygiene and needlestick injuries “. The question remains why 2 questions were chosen by the Authors to assess knowledge about hand hygiene and 2 others about theneedlestick injuries. Knowledge questions seem to be oriented to different issues. Such an instrument is not applicable for the purpose of this study and may lead to erroneous conclusions. I would suggest to delete knowledge-related issues from the analyses.

 The aim of the work was to learn about the behavior and attitudes  of medical personnel, using several socio-demographic variables and the level of knowledge. Selected questions regarding knowledge (and other questions) were analyzed by us in several works [i.e.:  Garus-Pakowska, A; Górajski, M; Szatko, F.: Awareness of the Risk of Exposure to Infectious Material and the Behaviors of Polish Paramedics with Respect to the Hazards from Blood-Borne Pathogens-A Nationwide Study. Int. J. Environ. Res. Public Health 2017 : V. 14, no. 8, pii. E843; DOI: 10.3390/ijerph14080843; or  Garus-Pakowska, A; Górajski, M; Szatko, F.: Knowledge and Attitudes of Dentists with Respect to the Risks of Blood-Borne Pathogens-A Cross-Sectional Study in Poland. Int. J. Environ. Res. Public Health 2017 : V. 14, no. 1, p. pii: E69; DOI: 10.3390/ijerph14010069]

and were intended to check the notion of hygienic behavior of physicians in general (not only NSIs).

The questions were intended to check the general concept of hygienic behavior of physicians. Hygiene and hand disinfection minimize the risk of transmission of microorganisms (as mentioned in the introduction), so we thought that questions are needed. At the same time, if not everybody knew the answer to such easy questions, we guess that with more difficult questions would be an even bigger problem. In Poland, it is estimated that over 60% of all HBV cases are nosocomial infections as a result of not maintaining universal precautions. That is why we also considered this question legitimate. We wanted to try to assess whether and how the level of knowledge about hygiene can affect, for example, the attitude of staff towards infected patients. However, we agree with the Reviewer that the test of knowledge was not a precise measure as we mentioned in the limitations section. However, we would like to once again point out that knowledge was not the purpose of the research (then a questionnaire should only be created to check knowledge), and knowledge was one of the variables used, the meaning of which we wanted to check. Moreover, if the respondents were not able to answer a few simple questions, we state that the level of knowledge is insufficient and that constant hygiene training and protection against NSIs should be carried out.

-          The explanation: “hospitals were located in urban communes and in urban-rural communes. In Poland, the commune is the basic unit of local self-government. The urban commune is a community located in a city. The urban-rural commune consists of a city that is the seat of the commune authorities and the surrounding villages” is confusing and is not related to the type of the hospital.

 “In the present analysis, the following sociodemographic measures were taken into account: gender, job category, work experience (in years), and place of employment (urban commune or rural and urban commune).”  Not all readers need to know the administrative division of Poland, therefore we have explained what these concepts mean. This is not the "category" of the hospital, but the location of the hospital. Location of the hospital / distance from the place where antiretroviral drugs are available may affect the attitude of the staff, their concerns, etc. That's why we made such a division.

3.     The Results section

-          Table 4 is not informative and is messy. It is not clear why some variables were taken separately e.g. “nurses” and “paramedics”, some others were  not, some are described in details e.g. “gender (male)”, some others are not e.g. “Work experience”. The authors should keep the same approach regarding all variables used in the analysis.

We have changed the presentation of variables in table 4 as suggested.

-          The variable “knowledge” should be deleted from the logistic regression analysis. This aspect is explained above.